# Domain Invariant Representation Learning with Domain Density Transformations

**A. Tuan Nguyen**
University of Oxford; VinAI Research
Oxford, United Kingdom
tuan@robots.ox.ac.uk

**Toan Tran**
VinAI Research
Hanoi, Vietnam
v.toantm3@vinai.io

**Yarin Gal**
University of Oxford
Oxford, United Kingdom
yarin@cs.ox.ac.uk

**Atilim Gunes Baydin**
University of Oxford
Oxford, United Kingdom
gunes@robots.ox.ac.uk

## Abstract

Domain generalization refers to the problem where we aim to train a model on data from a set of source domains so that the model can generalize to unseen target domains. Naively training a model on the aggregate set of data (pooled from all source domains) has been shown to perform suboptimally, since the information learned by that model might be domain-specific and generalize imperfectly to target domains. To tackle this problem, a predominant domain generalization approach is to learn some domain-invariant information for the prediction task, aiming at a good generalization across domains. In this paper, we propose a theoretically grounded method to learn a domain-invariant representation by enforcing the representation network to be invariant under all transformation functions among domains. We next introduce the use of generative adversarial networks to learn such domain transformations in a possible implementation of our method in practice. We demonstrate the effectiveness of our method on several widely used datasets for the domain generalization problem, on all of which we achieve competitive results with state-of-the-art models.

## 1 Introduction

Domain generalization refers to the machine learning scenario where the model is trained on multiple source domains so that it is expected to generalize well to unseen target domains. The key difference between domain generalization [25, 37, 18] and domain adaptation [49, 48, 14, 45] is that, in domain generalization, the learner does not have access to data of the target domain, making the problem much more challenging. One of the most common domain generalization approaches is to learn an invariant representation across domains, aiming at a good generalization performance on target domains. For instance, in the representation learning framework, the prediction function $y = f(x)$, where $x$ is data and $y$ is a label, is obtained as a composition $y = h \circ g(x)$ of a deep representation network $z = g(x)$, where $z$ is a learned representation of data $x$, and a smaller classifier $y = h(z)$, predicting label $y$ given representation $z$, both of which are shared across domains. With this framework, we can aim to learn an "invariant" representation $z$ across the source domains with the "hope" of a better generalization to the target domain.

Most existing "domain-invariance"-based methods in domain generalization focus on the marginal distribution alignment [37, 1, 44, 43, 32], which are still prone to distributional shifts when the conditional data distribution is not stable. In particular, the marginal alignment refers to making the

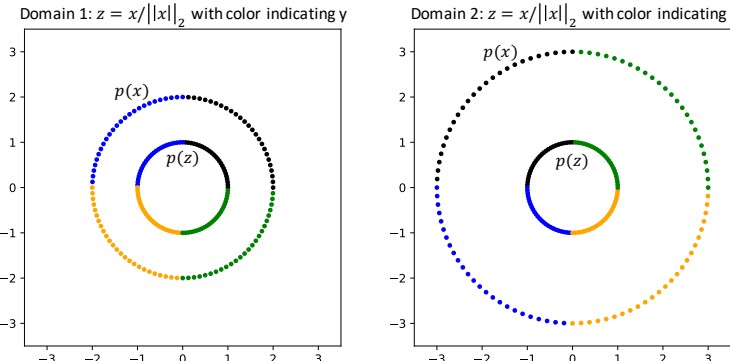

Figure 1: **An example of two domains**. For each domain, $x$ is uniformly distributed on the outer circle (radius 2 for domain 1 and radius 3 for domain 2), with the color indicating class label $y$. After the transformation $z = x/||x||_2$, the marginal of $z$ is aligned (uniformly distributed on the unit circle for both domains), but the conditional $p(y|z)$ is not aligned. Thus, using this representation for predicting $y$ would not generalize well across domains.

representation distribution $p(z)$ to be the same across domains. This is essential since if $p(z)$ for the target domain is different from that of source domains, the classification network $h(z)$ would face out-of-distribution data at test time. Conditional alignment refers to aligning the conditional distribution of the label given the representation $p(y|z)$ to expect that the classification network (trained on the source domains) would give accurate predictions at test time. The formal definitions of these two types of alignment are discussed in Section 3.

In Figure 1 we illustrate an example where the representation $z$ satisfies the marginal alignment but not the conditional alignment. Specifically, $x$ is distributed uniformly on the circle with radius 2 (and centered at the origin) for domain 1 and distributed uniformly on the circle with radius 3 (centered at the origin) for domain 2. The representation $z$ defined by the mapping $z = g(x) = x/||x||_2$ will align the marginal distribution $p(z)$, i.e., $z$ is now distributed uniformly on the unit circle for both domains. However, the conditional distribution $p(y|z)$ is not aligned between the two domains ($y$ is represented by color), which means using this representation for classification is suboptimal, and in this extreme case would lead to 0% accuracy in the target domain 2. This is an extreme case of misalignment but it does illustrate the importance of the conditional alignment. Therefore, we need to align both the marginal and the conditional distributions for a domain-invariant representation.

Recently, there have been several attempts [33, 34, 50] to align the joint distribution of the representation and the label $p(y, z)$ in a domain generalization problem by aligning the distribution of $z$ across domains for each class, i.e., $p(z|y)$ (given that the label distribution $p(y)$ is unchanged across domains). However, the key drawbacks of these methods are that they either do not scale well with the number of classes or have limited performance in real-world computer vision datasets (see Section 5).

In this paper, we focus on learning a domain-invariant representation that aligns both the marginal and the conditional distributions in domain generalization problems. We present theoretical results regarding the necessary and sufficient conditions for the existence of a domain-invariant representation; and subsequently propose a method to learn such representations by enforcing the invariance of the representation network under domain density transformation functions. A simple intuition for our approach is that if we enforce the representation to be invariant under the transformations among the source domains, the representation will become more robust under other domain transformations.

Furthermore, we introduce an implementation of our method in practice, in which the domain transformation functions are learned through the training process of generative adversarial networks (GANs) [20, 12]. We conduct extensive experiments on several widely used datasets and observe a significant improvement over relevant baselines. We also compare our methods against other state-of-the-art models and show that our method achieves competitive results.

Our contribution in this work is threefold:

- We revisit the domain invariant representation learning problem and shed some light by providing several observations: a necessary and sufficient condition for the existence of a

domain-invariant representation and a connection between domain-independent representation and a marginally-aligned representation.

- We propose a theoretically grounded method for learning a domain-invariant representation based on domain density transformation functions. We also demonstrate that we can learn the domain transformation functions by GANs in order to implement our approach in practice.

- We empirically show the effectiveness of our method by performing experiments on widely used domain generalization datasets (e.g., Rotated MNIST, VLCS and PACS) and compare our method with relevant baselines (especially CIDG [33], CIDDG [34] and DGER [50]).

## 2   Related Work

**Domain generalization:**   Domain generalization is an seminal task in real-world machine learning problems where the data distribution of a target domain might vary from that of the training source domains. Therefore, extensive research has been developed to handle that domain-shift problem, aiming at a better generalization performance in the unseen target domain. A predominant approach for domain generalization is domain invariance [37, 33, 34, 3, 47, 2, 24, 50, 1, 32, 44, 43] that learns a domain-invariant representation (which we define as to align the marginal distribution of the representation or the conditional distribution of the output given the representation or both). We are particularly interested in CIDG [33], CIDDG [34] and DGER [50], which also learn a representation that aligns the joint distribution of the representation and the label given that the class distribution is unchanged across domains. It should be noted that Zhao et al. [50] assume the label is distributed uniformly on all domains, while our proposed method only requires an assumption that the distribution of label is unchanged across domains (and not necessarily uniform). We also show later in our paper that the invariance of the distribution of class label across domains turns out to be the necessary and sufficient condition for the existence of a domain-invariant representation. Moreover, we provide a unified theoretical discussion about the two types of alignment, and then propose a method to learn a representation that aligns both the marginal and conditional distributions via domain density transformation functions for the domain generalization problem. Note that there exist several related works, such as Ajakan et al. [1], Ganin et al. [17], that use adversarial loss with a domain discriminator to align the marginal distribution of representation among domains, but they are different from our approach. In particular, our method only uses GANs or normalizing flows to learn the transformation among domains, and learn a representation that is invariant under these functions, without using an adversarial loss on the representation (which can lead to very unstable training [19, 27]). There also exist works [35, 23, 7, 40] in the domain adaptation literature that use generative modeling to learn a domain transformation function from source to target images, and use the transformed images to train a classifier. Our method differs from these by enforcing the representation to be invariant under the domain transformation, and we show theoretically that the representation learned that way would be domain-invariant marginally and conditionally. Meanwhile, the above works use the domain transformation to transform the images and train the classifier directly on the transformed data, and are not effective or applicable for domain generalization.

Another line of methods that received a recent surge in interest is applying the idea of meta-learning for domain generalization problems [16, 4, 31, 5]. The core idea behind these works is that if we train a model that can adapt among the source domains well, it would be more likely to adapt to unseen target domains. Recently, there are approaches [15, 9, 41] that make use of the domain specificity, together with domain invariance, for the prediction problem. The argument here is that domain invariance, while being generalized well between domains, might be insufficient for the prediction of each specific domain and thus domain specificity is necessary. We would like to emphasize that our method is not a direct competitor of meta-learning based and domain-specificity based methods. In fact, we expect that our method can be used in conjunction with these methods to get the best of both worlds for better performance.

**Density transformation between domains:**   Since our method is based on domain density transformations, we will review briefly some related works here. To transform the data density between domains, one can use several types of generative models. Two common methods are based on GANs [51, 12, 13] and normalizing flows [21]. Although our method is not limited to the choice of the generative model used for learning the domain transformation functions, we opt to use GAN,

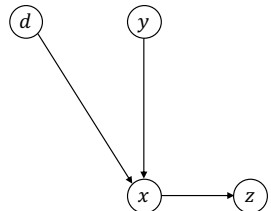

Figure 2: **Graphical model**. The data distribution is $p(x, y|d)$ for each domain $d$. Our goal is to learn a representation $z$ with a mapping $p(z|x)$ from $x$ so that $z$ can be generalized across domains for the prediction task.

specifically StarGAN [12], for scalability. This is just an implementation choice to demonstrate the use and effectiveness of our method in practice, and it is unrelated to our theoretical results.

**Connection to contrastive learning:**   Our method can be interpreted intuitively as a way to learn a representation network that is invariant (robust) under domain transformation functions. On the other hand, contrastive learning [10, 11, 36] is also a representation learning paradigm where the model learns images' similarity. In particular, contrastive learning encourages the representation of an input to be similar under different transformations (usually image augmentations). However, the transformations in contrastive learning are not learned and do not serve the purpose of making the representation robust under domain transformations. Our method first learns the transformations between domains and then uses them to learn a representation that is invariant under domain shifts.

## 3   Theoretical Approach

### 3.1   Problem Statement

Let us denote the data distribution for a domain $d \in \mathcal{D}$ by $p(x, y|d)$, where the variable $x \in \mathcal{X}$ represents the data and $y \in \mathcal{Y}$ is its corresponding label. The graphical model for our domain generalization framework is depicted in Figure 2, in which the joint distribution is presented as follows:

$$p(d, x, y, z) = p(d)p(y)p(x|y, d)p(z|x) . \tag{1}$$

In the domain generalization problem, since the data distribution $p(x, y|d)$ varies between domains, we expect the changes in the marginal data distribution $p(x|d)$ or the conditional data distribution $p(y|x, d)$ or both. In this paper, we assume that $p(y|d)$ is invariant across domains, i.e., the marginal distribution of the label $y$ is not dependent on the domain $d$—this assumption is shown to be the key condition for the existence of a domain-invariant representation (see Remark 1). This is practically reasonable since in many classification datasets, the class distribution can be assumed to be unchanged across domains (usually uniform distribution among the classes, e.g., balanced datasets).

Our aim is to find a domain-invariant representation $z$ represented by the mapping $p(z|x)$ that can be used for the classification of label $y$ and be generalized among domains. In practice, this mapping can be deterministic (in that case, $p(z|x) = \delta_{g_\theta(x)}(z)$ with some function $g_\theta$, where $\delta$ is the Dirac delta distribution) or probabilistic (e.g., a normal distribution with the mean and standard deviation outputted by a network parameterized by $\theta$). For all of our experiments, we use a deterministic mapping for an efficient inference at test time, while in this section, we present our theoretical results with the general case of a distribution $p(z|x)$.

In most existing domain generalization approaches, the domain-invariant representation $z$ is defined using one of the two following definitions:

**Definition 1.** *(Marginal Distribution Alignment) The representation $z$ is said to satisfy the marginal distribution alignment condition if $p(z|d)$ is invariant w.r.t. $d$.*

**Definition 2.** *(Conditional Distribution Alignment) The representation $z$ is said to satisfy the conditional distribution alignment condition if $p(y|z, d)$ is invariant w.r.t. $d$.*

However, when the joint data distribution varies between domains, it is crucial to align both the marginal and the conditional distribution of the representation $z$. To this end, this paper aims to

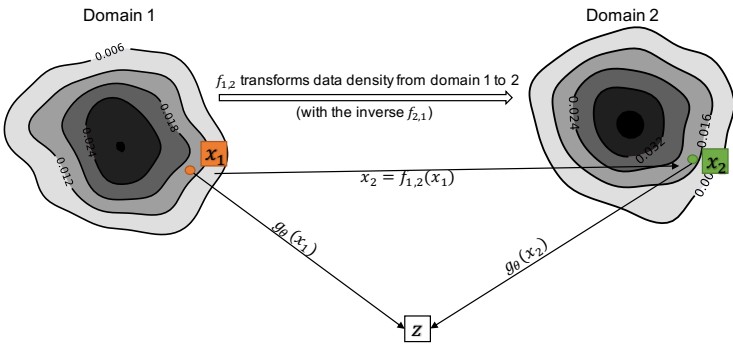

Figure 3: **Domain density transformation**. If we know the function $f_{1,2}$ that transforms the data density from domain 1 to domain 2, we can learn a domain invariant representation network $g_\theta(x)$ by enforcing it to be invariant under $f_{1,2}$, i.e., $g_\theta(x_1) = g_\theta(x_2)$ for any $x_2 = f_{1,2}(x_1)$ .

learn a representation $z$ that satisfies both the marginal and conditional alignment conditions. We justify our assumption of independence between $y$ and $d$ (thus $p(y|d) = p(y)$) by the following remark, which shows that this assumption turns out to be the necessary and sufficient condition for learning a domain-invariant representation. Note that this condition is also used in several existing works [50, 33, 34].

**Remark 1.** *The invariance of $p(y|d)$ across domains $d$ is the necessary and sufficient condition for the existence of a domain-invariant representation (that aligns both the marginal and conditional distributions).*

*Proof.* provided in the appendix. $\square$

It is also worth noting that methods which learn a domain independent representation, for example, [24], only align the marginal distribution. This comes directly from the following remark:

**Remark 2.** *A representation $z$ satisfies the marginal distribution alignment condition if and only if $I(z, d) = 0$, where $I(z, d)$ is the mutual information between $z$ and $d$.*

*Proof.* provided in the appendix. $\square$

The question still remains that how we can learn a non-trivial domain invariant representation that satisfies both of the distribution alignment conditions. This will be discussed in the following subsection.

## 3.2 Learning a Domain-Invariant Representation with Domain Density Transformation Functions

To present our method, we will make some assumptions about the data distribution. Specifically, for any two domains $d, d'$, we assume that there exists an invertible and differentiable function denoted by $f_{d,d'}$ that transforms the density $p(x|y, d)$ to $p(x'|y, d'), \forall y$. Let $f_{d,d'}$ be the inverse of $f_{d',d}$, i.e., $f_{d',d} := (f_{d,d'})^{-1}$.

Due to the invertibility and differentiability of $f$'s, we can apply the change of variables theorem [39, 6] for the distributions above. In particular, with $x' = f_{d,d'}(x)$ (and thus $x = f_{d',d}(x')$), we have:

$$p(x|y, d) = p(x'|y, d') \left| \det J_{f_{d',d}}(x') \right|^{-1}, \tag{2}$$

where $J_{f_{d',d}}(x')$ is the Jacobian matrix of the function $f_{d',d}$ evaluated at $x'$.

Multiplying both sides of Eq. 2 with $p(y|d) = p(y|d')$, we get

$$p(x, y|d) = p(x', y|d') \left| \det J_{f_{d',d}}(x') \right|^{-1}; \tag{3}$$

and marginalizing both sides of the above equation over $y$ gives us

$$p(x|d) = p(x'|d') \left| \det J_{f_{d',d}}(x') \right|^{-1}. \tag{4}$$

By using Eq. 2 and Eq. 4, we can prove the following theorem, which offers an efficient way to learn a domain-invariant representation, given the transformation functions $f$'s between domains.

**Theorem 1.** *Given an invertible and differentiable function $f_{d,d'}$ (with the inverse $f_{d',d}$) that transforms the data density from domain $d$ to $d'$ (as described above). Assuming that the representation $z$ satisfies:*

$$p(z|x) = p(z|f_{d,d'}(x)), \ \forall x, \tag{5}$$

*Then it aligns both the marginal and the conditional of the data distribution for domain $d$ and $d'$.*

*Proof.* provided in the appendix. $\square$

This theorem indicates that, if we can find the functions $f$ that transform the data densities among the domains, we can learn a domain-invariant representation $z$ by encouraging the representation to be invariant under all the transformations $f$. This idea is illustrated in Figure 3. We therefore can use the following learning objective to learn a domain-invariant representation $z = g_\theta(x)$:

$$\mathbb{E}_d \left[ \mathbb{E}_{p(x,y|d)} \left[ l(y, g_\theta(x)) + \mathbb{E}_{d'}[dis(g_\theta(x), g_\theta(f_{d,d'}(x)))] \right] \right] \tag{6}$$

Assume that we have a set of $K$ sources domain $D_s = \{d_1, d_2, ..., d_K\}$, the objective function in Eq. 6 becomes:

$$\mathbb{E}_{d,d' \in D_s, p(x,y|d)} \left[ l(y, g_\theta(x)) + dis(g_\theta(x), g_\theta(f_{d,d'}(x))) \right], \tag{7}$$

where $l(y, g_\theta(x))$ is the prediction loss of a network that predicts $y$ given $z = g_\theta(x)$, and $dis$ is a distance metric to enforce the invariant condition in Eq. 5. In our implementation, we use a squared error distance, e.g., $dis(g_\theta(x), g_\theta(f_{d,d'}(x))) = ||g_\theta(x) - g_\theta(f_{d,d'}(x))||_2^2$, since it performs the best in practice. However, we also considered other distances such as constrastive distance, which we discuss in more detail in the appendix.

This theorem motivates us to learn such domain transformation functions for our domain-invariant representation learning framework. In the next section, we show how one can incorporate this idea into real-world domain generalization problems by learning the transformations with generative adversarial networks.

## 4 An Practical Implementation using Generative Adversarial Networks

In practice, we can learn the functions $f$ that transform the data distributions between domains by using several generative modeling frameworks, e.g., normalizing flows [21] or GANs [51, 12, 13]. One advantage of normalizing flows is that this transformation is naturally invertible by design of the neural network. However, existing frameworks (e.g., Grover et al. [21]) require two flows to transform between each pair of domains, making it not scalable (scales linearly with the number of domains). Moreover, an initial implementation of our method using AlignFlow shows similar performance with the version using GAN. Therefore, we opt to use GANs for better scalability. In particular, we use the StarGAN [12] model, which is a unified network (only requiring a single network to transform across all domains) designed for image domain transformations. It should be noted that the transformations learned by StarGAN are differentiable everywhere or almost everywhere with typical choices of the activation function (e.g., tanh or ReLU), and the cycle-consistency loss enforces each pair of transformations to approximate a pair of inverse functions.

The goal of StarGAN is to learn a unified network $G$ that transforms the data density among multiple domains. In particular, the network $G(x, d, d')$ (i.e., $G$ is conditioned on the image $x$ and the two different domains $d, d'$) transforms an image $x$ from domain $d$ to domain $d'$. Different from the original StarGAN model that only takes the image $x$ and the desired destination domain $d'$ as its input, in our implementation, we feed both the original domain $d$ and desired destination domain $d'$ together with the original image $x$ to the generator $G$.

The generator's goal is to fool a discriminator $D$ into thinking that the transformed image belongs to the destination domain $d'$. In other words, the equilibrium state of StarGAN, in which $G$ completely fools $D$, is when $G$ successfully transforms the data density of the original domain to that of the destination domain. After training, we use $G(., d, d')$ as the function $f_{d,d'}(.)$ described in the previous section and perform the representation learning via the objective function in Eq. 7.

Three important loss functions of the StarGAN architecture are:

- Domain classification loss $\mathcal{L}_{cls}$ that encourages the generator $G$ to generate images that closely belongs to the desired destination domain $d'$.
- The adversarial loss $\mathcal{L}_{adv}$ that is the classification loss of a discriminator $D$ that tries to distinguish between real images and the synthetic images generated by G. The equilibrium state of StarGAN is when $G$ completely fools $D$, which means the distribution of the generated images (via $G(x, d, d'), x \sim p(x|d)$) becomes the distribution of the real images of the destination domain $p(x'|d')$. This is our objective, i.e., to learn a function that transforms domains' densities.
- Reconstruction loss $\mathcal{L}_{rec} = \mathbb{E}_{x,d,d'}[||x - G(x', d', d)||_1]$ where $x' = G(x, d, d')$ to ensure that the transformations preserve the image's content. Note that this also aligns with our interest since we want $G(., d', d)$ to be the inverse of $G(., d, d')$, which minimizes $\mathcal{L}_{rec}$.

We can enforce the generator $G$ to transform the data distribution within the class $y$ (e.g., $p(x|y, d)$ to $p(x'|y, d') \; \forall y$) by sampling each minibatch with data from the same class $y$, so that the discriminator will distinguish the transformed images with the real images from class $y$ and domain $d'$. However, we found that this constraint can be relaxed in practice, and the generator almost always transforms the image within the original class $y$.

As mentioned earlier, after training the StarGAN model, we can use the generator $G(., d, d')$ as our $f_{d,d'}(.)$ function and learn a domain-invariant representation via the learning objective in Eq. 7. We name this implementation of our method DIRT-GAN (Domain Invariant Representation learning with domain Transformations via Generative Adversarial Networks).

## 5 Experiments

### 5.1 Datasets

To evaluate our method, we perform experiments in three datasets that are commonly used in the literature for domain generalization.

**Rotated MNIST [18]:**    In this dataset, 1,000 MNIST images (100 per class) [29] are chosen to form the first domain (denoted $\mathcal{M}_0$), then counter-clockwise rotations of $15°, 30°, 45°, 60°$ and $75°$ are applied to create five additional domains, denoted $\mathcal{M}_{15}, \mathcal{M}_{30}, \mathcal{M}_{45}, \mathcal{M}_{60}$ and $\mathcal{M}_{75}$. The task is classification with ten classes (digits 0 to 9).

**VLCS [18]:**    contains 10,729 images from four domains, each domain is a subdataset. The four datasets are VOC2007 (V), LabelMe (L), Caltech-101 (C), and SUN09 (S). The task is classification with five classes.

**PACS [30]:**    contains 9,991 images from four different domains: art painting, cartoon, photo, sketch. The task is classification with seven classes.

### 5.2 Experimental Setting

For all datasets, we perform "leave-one-domain-out" experiments, where we choose one domain as the target domain, train the model on all remaining domains and evaluate it on the chosen domain. Following standard practice, we use 90% of available data as training data and 10% as validation data, except for the Rotated MNIST experiment where we do not use a validation set and just report the performance of the last epoch.

For the **Rotated MNIST** dataset, we use a network of two 3x3 convolutional layers and a fully connected layer as the representation network $g_\theta$ to get a representation $z$ of 64 dimensions. A single

Table 1: **Rotated Mnist**. Reported numbers are mean accuracy and standard deviation among 5 runs

| Model | Domains | | | | | | Average |
|---|---|---|---|---|---|---|---|
| | $\mathcal{M}_0$ | $\mathcal{M}_{15}$ | $\mathcal{M}_{30}$ | $\mathcal{M}_{45}$ | $\mathcal{M}_{60}$ | $\mathcal{M}_{75}$ | |
| HIR [47] | 90.34 | 99.75 | 99.40 | 96.17 | 99.25 | 91.26 | 96.03 |
| DIVA [24] | 93.5 | 99.3 | 99.1 | 99.2 | 99.3 | 93.0 | 97.2 |
| DGER [50] | 90.09 | 99.24 | 99.27 | 99.31 | 99.45 | 90.81 | 96.36 |
| DA [17] | 86.7 | 98.0 | 97.8 | 97.4 | 96.9 | 89.1 | 94.3 |
| LG [42] | 89.7 | 97.8 | 98.0 | 97.1 | 96.6 | 92.1 | 95.3 |
| HEX [46] | 90.1 | 98.9 | 98.9 | 98.8 | 98.3 | 90.0 | 95.8 |
| ADV [46] | 89.9 | 98.6 | 98.8 | 98.7 | 98.6 | 90.4 | 95.2 |
| DIRT-GAN (ours) | 97.2($\pm$0.3) | 99.4($\pm$0.1) | 99.3($\pm$0.1) | 99.3($\pm$0.1) | 99.2($\pm$0.1) | 97.1($\pm$0.3) | **98.6** |

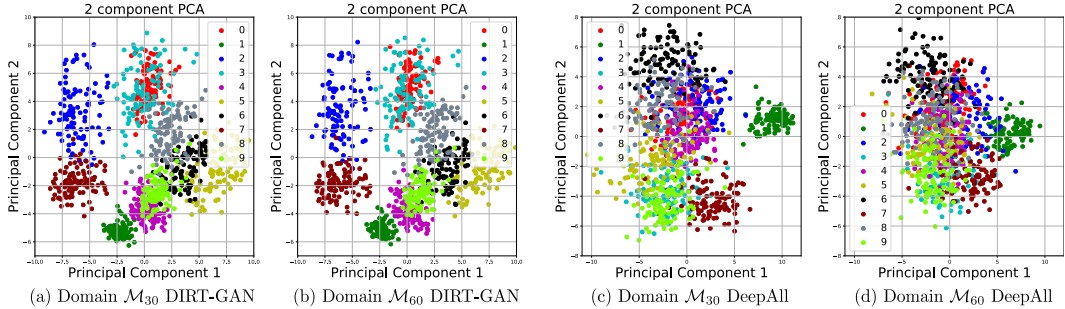

(a) Domain $\mathcal{M}_{30}$ DIRT-GAN    (b) Domain $\mathcal{M}_{60}$ DIRT-GAN    (c) Domain $\mathcal{M}_{30}$ DeepAll    (d) Domain $\mathcal{M}_{60}$ DeepAll

Figure 4: **Visualization of the representation space**. Each point indicates a representation $z$ of an image $x$ in the two dimensional space and its color indicates the label $y$. Two left figures are for our method DIRT-GAN and two right figures are for the naive model DeepAll.

linear layer is then used to map the representation $z$ to the ten output classes. This architecture is the deterministic version of the network used by Ilse et al. [24]. We train our network for 500 epochs with the Adam optimizer [26], using the learning rate 0.001 and minibatch size 64, and report performance on the test domain after the last epoch.

For the **VLCS** and **PACS** datasets, for a fair comparison against our main baselines, we use the most common choices of backbone networks for those datasets in existing works as the representation networks $g_\theta$, i.e., Alexnet [28] for VLCS and Resnet18 [22] for PACS. We replace the last fully connected layer of the backbone with a linear layer of dimension 256 so that our representation has 256 dimensions. As with the Rotated MNIST experiment, we use a single layer to map from the representation $z$ to the output. We train the network for 100 epochs with plain stochastic gradient descent (SGD) using learning rate 0.001, momentum 0.9, minibatch size 64, and weight decay 0.001. Data augmentation is also standard practice for real-world computer vision datasets like VLCS and PACS, and during the training we augment our data as follows: crops of random size and aspect ratio, resizing to 224 × 224 pixels, random horizontal flips, random color jitter, randomly converting the image tile to grayscale with 10% probability, and normalization using the ImageNet channel means and standard deviations.

The StarGAN [12] model implementation is taken from the authors' original source code with no significant modifications. For each set of source domains, we train the StarGAN model for 100,000 iterations with a minibatch of 16 images per iteration.

Our code is available at https://github.com/atuannguyen/DIRT. We train our model on a NVIDIA Quadro RTX 6000.

## 5.3 Results

**Rotated MNIST Experiment.** Table 1 shows the performance of our model on the Rotated MNIST dataset. The main baselines we consider in this experiment are HIR [47], DIVA [24] and DGER [50], which are domain invariance based methods. Our method recognizably outperforms those, illustrating

Table 2: **VLCS**. Reported numbers are mean accuracy and standard deviation among 5 runs

| Model | Backbone | VLCS | | | | |
| | | V | L | C | S | Average |
|---|---|---|---|---|---|---|
| CIDG [33] | Alexnet | 65.65 | 60.43 | 91.12 | 60.85 | 69.51 |
| CIDDG [34] | Alexnet | 64.38 | 63.06 | 88.83 | 62.10 | 69.59 |
| DGER [50] | Alexnet | 73.24 | 58.26 | 96.92 | 69.10 | 74.38 |
| HIR [47] | Alexnet | 69.10 | 62.22 | 95.39 | 65.71 | 73.10 |
| JiGen [8] | Alexnet | 70.62 | 60.90 | 96.93 | 64.30 | 73.19 |
| DIRT-GAN (ours) | Alexnet | 72.1($\pm$1.0) | 64.0($\pm$0.9) | 97.3($\pm$0.2) | 72.2($\pm$1.1) | **76.4** |

Table 3: **PACS**. Reported numbers are mean accuracy and standard deviation among 5 runs

| Model | Backbone | PACS | | | | |
| | | Art Painting | Cartoon | Photo | Sketch | Average |
|---|---|---|---|---|---|---|
| DGER [50] | Resnet18 | 80.70 | 76.40 | 96.65 | 71.77 | 81.38 |
| JiGen [8] | Resnet18 | 79.42 | 75.25 | 96.03 | 71.35 | 79.14 |
| MLDG [31] | Resnet18 | 79.50 | 77.30 | 94.30 | 71.50 | 80.70 |
| MetaReg [4] | Resnet18 | 83.70 | 77.20 | 95.50 | 70.40 | 81.70 |
| CSD [38] | Resnet18 | 78.90 | 75.80 | 94.10 | 76.70 | 81.40 |
| DMG [9] | Resnet18 | 76.90 | 80.38 | 93.35 | 75.21 | 81.46 |
| DIRT-GAN (ours) | Resnet18 | 82.56($\pm$ 0.4) | 76.37($\pm$ 0.3) | 95.65($\pm$ 0.5) | 79.89($\pm$ 0.2) | **83.62** |

the effectiveness of our method in learning a domain-invariant representation over the existing works. We also include other best-performing models for this dataset in the second half of the table. To the best of our knowledge, we set a new state-of-the-art performance on this Rotated MNIST dataset.

We further analyze the distribution of the representation $z$ by performing principal component analysis to reduce the dimension of $z$ from 64 to two principal components. We visualize the representation space for two domains $\mathcal{M}_{30}$ and $\mathcal{M}_{60}$, with each point indicating the representation $z$ of an image $x$ in the two-dimensional space and its color indicating the label $y$. Figures 4a and 4b show the representation space of our method (in domains $\mathcal{M}_{30}$ and $\mathcal{M}_{60}$ respectively). It is clear that both the marginal (judged by the general distribution of the points) and the conditional (judged by the positions of colors) are relatively aligned. Meanwhile, Figures 4c and 4d show the representation space with naive training (in domains $\mathcal{M}_{30}$ and $\mathcal{M}_{60}$ respectively), showing the misalignment in the marginal distribution (judged by the general distribution of the points) and the conditional distribution (for example, the distributions of blue points and green points).

**VLCS and PACS.** Tables 2 and 3 show the results for the VLCS and PACS datasets. In these real-world computer vision datasets, we consider HIR [47], CIDG [33], CIDDG [34] and DGER [50] as our main domain-invariance baselines. We also include other approaches (meta-learning based or domain-specificity based) in the second half of the tables for references. Our method significantly ourperforms other invariant-representataion baselines, namely CIDG, CIDDG and DGER, with the same backbone architectures, showing the effectiveness of our representation alignment method.

## 6   Conclusion

To conclude, in this work we propose a theoretically grounded approach to learn a domain-invariant representation for the domain generalization problem by using domain transformation functions. We also provide insights into domain-invariant representation learning with several theoretical observations. We then introduce an implementation for our method in practice with the domain transformations learned by a StarGAN architecture and empirically show that our approach outperforms other domain-invariance based methods. Our method also achieves competitive results on several datasets when compared to other state-of-the-art models. A potential limitation of our method is that we need to train an additional network (StarGAN) to learn to transform data density among domains.

However, this network is only used during training, and the required computation at test time is still the same as other models. In the future, we plan to incorporate our method into meta-learning based and domain-specificity based approaches for improved performance. We also plan to extend the domain-invariant representation learning framework to the more challenging scenarios, for example, where domain information is not available (i.e., we have a dataset pooled from multiple source domains but do not know the domain identification of each data instance).

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
