# OpenReview forum: "Domain Invariant Representation Learning with Domain Density Transformations"
_NeurIPS.cc/2021/Conference — NeurIPS 2021 Poster_

### Official Review · Reviewer_TszG · 2021-07-17

**Rating:** 4
**Confidence:** 4

**Summary:**

The paper presents a method for domain generalization in the presence of multiple source domains.  Given a pre-trained domain translation model f, the proposed method regularizes ERM by minimizing the l2 distance between  g(x) and its translation in domain d'; g(f_d'(x)) , where g is the feature extractor.

**Limitations And Societal Impact:**

The authors briefly talk about the limitation of their work as having to train an additional domain translation method (STARGAN).
Another potential limitation of the model could be its ability to deal with label distribution shift. An experiment is needed to verify if that is indeed the case.

**Main Review:**

Strengths of the paper:
-------------------------------
The idea of using domain translation as an intervention to learn invariant features is interesting. I think this idea could be expanded to include arbitrary transformations for arbitrary domains.

Weaknesses of the paper:
-----------------------------------
- The authors designate a complete page on using invertible function for domain translation but finally opt for stargan. I don't believe stargan learns invertible functions.
- shouldn't  the goal is to learn invariant functions for p(y|g(x))? I.e for any domain transformation f, p(y|g(x) and p(y|g(f(x))) should be the same. In other words, why is the regularization done in feature space? A better discussion on this is needed.
- since we do have access to labels in the source domain, what would happen if we just minimize “dist(g(x_{d1}|y=c), g(x_{d2}|y=c))” where d1 and d2 represent domain 1 and 2. Probably this is a big regularization on the model since this way the model would learn common information in x_d1 and x_d2 which relate to class c and this may not generalize well to other domains but an experiment like this could act as an ablation study and highlight the merits of the proposed approach.
- the paper lacks comparison with IRM and IRM based methods.




**Time Spent Reviewing:**

5 hours

---

> ### Author Response · Authors · 2021-08-10
> **Rebuttal**
>
> We thank the reviewer for helping to review the paper. We address your concerns below:
>
> **The authors designate a complete page on using invertible function for domain translation but finally opt for stargan. I don't believe stargan learns invertible functions.**
>
> This is correct that the function learned by StarGAN is generally not invertible. In the preliminary experiments, we have both the implementation with StarGAN and AlignFlow (which is invertible). However, we observed similar performance from the two versions. Therefore, we opt to use StarGAN for its scalability (see discussion of this in lines 212-222). We conjecture that this is because StarGAN also tries to approximate pairs of inverse functions due to its cycle consistency loss (although they are not actually invertible). We will include both versions/implementations in the revised version of our paper. Also, note that our method is not limited to the choice of the transformation network. We can use AlignFlow (which is guaranteed to be invertible), or any other available models in the literature. StarGAN is just an implementation choice that works well in practice, and is not part of our theoretical method discussed in section 3.
>
> **shouldn't the goal is to learn invariant functions for p(y|g(x))? I.e for any domain transformation f, p(y|g(x) and p(y|g(f(x))) should be the same. In other words, why is the regularization done in feature space? A better discussion on this is needed.**
>
> Yes, the final goal is to learn invariant representation z that p(y|z) and p(z) do not change across domains. But learning the representation this way is difficult, since it is hard to compute or approximate p(z) and p(y|z). However, Theorem 1 shows that we can learn such a representation by forcing invariance under the domain transformations in the representation space directly. This is the motivation of our method.
>
> **since we do have access to labels in the source domain, what would happen if we just minimize “dist(g(x_{d1}|y=c), g(x_{d2}|y=c))” where d1 and d2 represent domain 1 and 2. Probably this is a big regularization on the model since this way the model would learn common information in x_d1 and x_d2 which relate to class c and this may not generalize well to other domains but an experiment like this could act as an ablation study and highlight the merits of the proposed approach.**
>
> This is indeed the idea behind the baselines CIDG and CIDDG. However, a limitation of this idea is that it does not scale well with the number of classes (need to align the distribution for each class and each pair of domains, and is very computationally expensive when the number of classes is large). Also, these methods have not achieved good performance on real-world computer vision datasets. Table 2 shows that our method clearly outperforms them.
>
> **the paper lacks comparison with IRM and IRM based methods.**
>
> When conducting the experiments, we intended to only include baselines that report the results in these datasets for a fair comparison (since authors usually do the most proper and extensive hyper-parameter tuning for their model). This is also the reason that we use Alexnet for VLCS and ResNet18 for PACS, the same as our main baselines. During the rebuttal period, we run the experiments for IRM in PACS with ResNet18 using the code of DomainBed ([1]) and their hyper-parameter tuning strategy. The results are as follow:
>
> | Methods    | A | C | P | S | Average |
> |  :-:  |  :-:  |  :-:  |  :-:  |  :-:  |  :-:  |
> | IRM        | 75.1±1.3 | 71.0±0.6 | 93.8±0.3 | 71.2±3.4 | 77.8 |
> | DIRT-GAN (ours)  | **82.56±0.4** | **76.37±0.3** | **95.65±0.5** | **79.89±0.2** | **83.62** |
>
> Our model outperforms IRM by quite a large margin.
>
> [1] Gulrajani, Ishaan, and David Lopez-Paz. "In search of lost domain generalization." arXiv preprint arXiv:2007.01434 (2020).

---

> > ### Comment · Reviewer_TszG · 2021-08-17
> > **the importance of the domain translator  model?**
> >
> > I would like to thank the authors for providing more baselines. The performance of stargan or any other domain translation model seems to be the key in the proposed approach.  I wonder how dependant their model is to stargan. Can the authors show the generalizability of the approach using other domain translation models.
> > Also We know that gan based approaches are vulnerable to class distribution shift between domains. How does the proposed model perform under class distribution shift between domains?

---

> > > ### Author Response · Authors · 2021-08-18
> > > **The model's performance is not dependant to the choice of domain transformation model**
> > >
> > > Thanks for your question. As mentioned in the rebuttal, the model's performance is not too sensitive or dependant on the choice of the domain transformation model. In the preliminary experiments, we had both the implementation with StarGAN and AlignFlow (normalizing flows based, and is invertible by design). We observed similar performance for both versions, detailed numbers for the PACS dataset are below:
> > >
> > > | Methods    | A | C | P | S | Average |
> > > |  :-:  |  :-:  |  :-:  |  :-:  |  :-:  |  :-:  |
> > > | DIRT-FLOW (ours) | 82.03±0.3 | 76.41±0.3 | 95.84±0.4 | 80.35±0.2 | 83.66 |
> > > | DIRT-GAN (ours)  | 82.56±0.4 | 76.37±0.3 | 95.65±0.5 | 79.89±0.2 | 83.62 |
> > >
> > > The table above shows that we achieve similar performance with AlignFlow or StarGAN, indicating that the model's performance is not dependant on the choice of the domain translation model. We only choose StarGAN for its scalability (compared to AlignFlow). We will make sure to include both implementation versions in the revised paper.

---

> > > ### Author Response · Authors · 2021-08-24
> > > **Regarding the concern about label shift problem**
> > >
> > > Dear reviewer,
> > >
> > > We also want to address your concern about the label shift problem.
> > >
> > > Our paper is based on the assumption that $p(y|d)$ doesn't change across domains (line 140). This means that we are already assuming that there is no label shift across domains. In remark 1 we prove that this assumption/condition is actually needed for all domain marginally- and conditionally- invariant representation methods (not just ours), and these methods also actually do use this assumption. In practice, our model (and other domain invariance methods) can still perform well in case of non-extreme label shifts (e.g., PACS, VLCS). Also, we **reemphasize** that our proposed method is not dependent on StarGAN, even if StarGAN struggle with non-balance datasets, we can use other domain translation methods (e.g., AlignFlow) instead.
> > >
> > > Nevertheless, we conduct some experiments to investigate the performance of the model in case of more extreme label shifts. We run the experiment in the PACS dataset with source domains A,C,P, and target domain S. The "artificially" create more extreme label shift among the source domains as below:
> > > - Domain A: upsample class 0 ten times in the data loader, which makes this domain imbalanced towards class 0.
> > > - Domain C: upsample class 1 ten times in the data loader.
> > > - Domain P: upsample class 2 ten times in the data loader.
> > >
> > > The performance of DIRT-GAN for the target domain S in this experiment is 78.95 (compared with 79.89 originally), which shows that our model is not very sensitive against label shift. We also check the accuracy of each class (this dataset has a total of 7 classes) for the target domain S, and we found out that the numbers are also similar for both setups.
> > >
> > > Also, note that the case of extreme label shift is a challenging problem for domain generalization in general; and that almost all DG methods usually do not perform best under this case. However, an "easy fix" to alleviate the problem is that **we can do class-weighted sampling for each source domain to eliminate the label shifts among source domains** - this strategy is used by DGER [1] (this was not used in the above experiment)
> > >
> > > [1] Zhao, Shanshan, et al. "Domain generalization via entropy regularization." Advances in Neural Information Processing Systems 33 (2020).

---

> > > ### Author Response · Authors · 2021-08-30
> > > **Discussion period's deadline is approaching**
> > >
> > > Dear reviewer,
> > >
> > > As the discussion period is nearing an end, we wonder if you can spend some time going over our newest replies to see if they have successfully addressed your concerns or not. Please let us know if you have any remaining/additional concerns.
> > >
> > > Thank you very much.
> > >
> > > Best regards,
> > >
> > > Authors.

---

> ### Comment · Reviewer_TszG · 2021-09-03
> **Improving my score**
>
> The authors have addressed a lot of my concerns. They provided more baselines and have clarified questions. After having read the authors responses I would like to improve my rating to 6.

---

> > ### Author Response · Authors · 2021-09-03
> > **Thank you**
> >
> > Dear reviewer,
> >
> > Thank you for your reconsideration.
> >
> > If it is not too much trouble to ask, can you also edit the score in the original review to avoid any confusion for the AC?
> >
> > Thank you very much.
> >
> > Best regards,
> >
> > Authors.

---

### Official Review · Reviewer_RKBf · 2021-07-17

**Rating:** 6
**Confidence:** 3

**Summary:**

The paper proposes a novel method, DRIT-GAN, to learn domain invariant representation for domain generalization.

**Limitations And Societal Impact:**

As mentioned, the paper has limited depth in theory. The author could try to think about theoretical analysis that bounding your model's error. For example, what if your StarGAN cannot perfectly learn the transformation? How its error of learning transformation translates to the error of your final prediction model?

**Main Review:**

Originality: The task of domain generalization (DG) is classic. The proposed method leverages the well-known technique, StarGAN. The novelty is limited.

Quality: Theoretically, I am afraid that the paper is not technically sound. Remarks 1 & 2 are trivial and immediate from the definition. The author claims several prior works used those conditions without noting them. However, I am not clear about the value of remarks 1 & 2 in developing the proposed method. Here are questions to the author:
1. You use remark 1 to justify your assumption that y and d are independent. So how does the assumption of y and d being independent guide your algorithm design? Will your method be broken if y and d are not independent?
2. By remark 2, the author tries to say that enforcing representation z to be independent of domain d is not enough since it only aligns the marginal. However, most prior methods about DG not only enforce 'z' and 'd' being independent. They also use z to predict y. So at least implicitly, they are aligning the conditional distribution. Does the author aware of it?

Then the author put a new assumption, existing invertible transformations between domains, and proposes a method that works under this assumption.
1. How do you validate your assumption? How likely it satisfies in actual practice?
2. I notice the author assumes that the transformations only depend on data x, domain d, but are independent of label y. Can you justify why it is independent of y?

The theorem says the proposed method is valid (correct), but it does not say it is optimal. Theorem gives a sufficient condition but not a necessary one. So I do curious about the optimality of the method. Let's consider the motivating example that the author presented in Figure 1. Does your method work in this case? Can StarGAN learn the correct transformation between two domains? I feel the StarGAN also has the risk of being trapped into the suboptimal as the author described in the introduction and ending with 0% accuracy.

Empirically, the author does a good job of showing performance improvement. Clearly, your method shows an advantage over prior works on three benchmarks. A deeper analysis would be good if the author can shed more light on why the proposed method improves the performance.

Clarity: The paper is well-written. As a reader, I can easily follow it.

**Time Spent Reviewing:**

10

---

> ### Author Response · Authors · 2021-08-10
> **Rebuttal**
>
> Dear reviewers,
>
> Thank you for your comment. We answer your questions below:
>
> **You use remark 1 to justify your assumption that y and d are independent. So how does the assumption of y and d being independent guide your algorithm design? Will your method be broken if y and d are not independent?**
>
> If y and d are not independent (i.e., there is label shift among domains), the theoretical guarantee and motivation of our method would be broken. In particular, theorem 1 would not hold if this assumption is violated (we use the assumption repeatedly to prove it, see line 187 and the proof in the appendix). What we show here is that this assumption is needed for all domain (marginally and conditionally) invariant representation methods. However, in practice, our model (and others) still performs well in the case of non-extreme label shift (for example, in the VLCS and PACS dataset, the distribution of label is not exactly equal across all domains).
>
> **By remark 2, the author tries to say that enforcing representation z to be independent of domain d is not enough since it only aligns the marginal. However, most prior methods about DG not only enforce 'z' and 'd' being independent. They also use z to predict y. So at least implicitly, they are aligning the conditional distribution. Does the author aware of it?**
>
> Yes, when learning z to predict y, the conditional distribution is implicitly aligned. This is also the reason that the traditional Empirical Risk Minimization baseline performs decently in many datasets ([1]). However, a great amount of research has shown that explicitly aligning both the marginal and conditional distribution leads to better generalization performance in many DG datasets (see the baselines CIDA, CIDDG, or DGER). Therefore, we argue that explicitly aligning the conditional distribution is important and can lead to great improvement in many cases, and domain-independent representation methods do not explicitly do that.
>
> **I notice the author assumes that the transformations only depend on data x, domain d, but are independent of label y. Can you justify why it is independent of y?**
>
> This is according to our theoretical analysis (discussed in section 3.2). As you can see there, the **same** function $f_{d,d'}(x)$ transforms $p(x|y,d)$ to $p(x'|y,d')$ **for all** y, so $y$ should not (and does not need to) be a part of that transformation. The intuition behind this is also very simple. For example, if we have 2 datasets of digit numbers (from 0 to 9) of 2 domains. A good transformation that understands the underlying dynamic of the datasets would transform images of class i of one domain to images of the **same** class i of the other domain, for all i. Furthermore, later we enforce the representation to be invariant under the transformation, meaning $g_\theta(x)=g_\theta(f(x))$; so if the function $f$ transform cross classes (e.g., from class 0 to class 1), we would force the representation of a digit 0 to be the same as the representation of a digit 1, which is problematic.
>
> **The theorem says the proposed method is valid (correct), but it does not say it is optimal.**
>
> We are confused as to what the reviewer means by "optimal"? Clearly, there are many domain-invariant representations, and several ways to learn one (our method and the baselines CIDDG, CIDG, DGER). How can we classify one as optimal? If by "optimal", the reviewer means that the representation contains the maximum information about the class label $y$, then it is handled by the first term of the objective - the task loss. The task loss (normally cross-entropy loss) is an upper bound of the negative mutual information between z and y $-I(z,y)$. Hence, by minimizing this loss, the model will try to find an invariant representation (within its search space) that contains the most information about the class label. See more discussion about this from ([2]). However, if by optimal, the reviewer means that if our method will be able to find ANY domain invariant representation, then the answer would be no. Currently, to the best of our knowledge, there are no methods that can do this (all current domain invariant representation methods just propose **a** way to learn **a** domain invariant representation).
>
> In the motivating example, StarGAN is able to find the correct transformation. The transformation is even pretty easy to find in this case: rotation by $90^o$ counter-clockwise followed by dilation with scale factor $\frac{3}{2}$. We only need to sample each minibatch with datapoints from a same class and condition the discriminator on the class (as discussed in lines 246-248).
>
>
> [1] Gulrajani, Ishaan, and David Lopez-Paz. "In search of lost domain generalization." arXiv preprint arXiv:2007.01434 (2020).
>
> [2] Alemi, Alexander A., et al. "Deep variational information bottleneck." arXiv preprint arXiv:1612.00410 (2016).

---

> > ### Comment · Reviewer_RKBf · 2021-08-23
> > **Thanks for the rebuttal. One remain question.**
> >
> > First, I appreciate the author's answers. Basically, given the assumption that the label "y" and domain "d" are independent, the author's arguments are valid. I will not push the author to discuss the setting where exists label shift.
> >
> > Second, as already mentioned by other reviewers, compare with other non-"domain invariance" methods of domain generation can make the experiment more solid. It seems the author already compared their method with IRM which is good.
> >
> > Third, I would like to explain what I mean by "The theorem says the proposed method is valid (correct), but it does not say it is optimal." I am sorry that I did not make it clear in the original review.
> > As I said, theorem 1 is a statement about sufficient conditions while not talking about the necessary conditions. Let me be specific. Theorem 1 basically says: model satisfied your condition is model desired. However, it does not guarantee that all models desired satisfied your conditions. So here comes the risk that what if all models desired (i.e. those aligning both the marginal and the conditional) do not satisfy your condition? Then will your method be sub-optimal, i.e. optimizing your loss can not deliver the best model we can get.
> >
> > Now let's come back to your motivating example. In my original review, I am saying StarGAN may get a 0% accuracy because I thought in the figure, one is source domain, one is target domain. While in the rebuttal, I find the author is considering both domains are source domains and claiming that StarGAN is able to make 100% accuracy. Which is fine. I agree that StarGAN can make it correct if both are source domains. However, I am still unconvinced that StarGAN can generalize to new target domains. Consider, now I have a new domain with a different radius and a different rotation angle. It is hard for me to believe that StarGAN can successfully rotate the data to align the target with the source perfectly. Does the author have any thoughts about whether StarGAN can successfully generalize and why?

---

> > > ### Author Response · Authors · 2021-08-24
> > > **Discussion (continued)**
> > >
> > > Dear reviewer,
> > >
> > > Thank you for engaging in the discussion to help improve the paper, and for the clarification.
> > >
> > > **Regarding non-"domain invariance" baselines**
> > >
> > > As the reviewer may have noticed, our method outperforms IRM by quite a large margin. In the main paper, we also tried to include some other non-"domain invariance" baselines such as JiGen, MLDG, MetaReg, CSD, DMG (please refer to tables 2 and 3). If the review has any other baseline suggestions or requests, please let us know. We will try our best to tune the hyper-parameters and run the experiments for that method in the remaining time of the discussion period.
> > >
> > > For the generalization of the model to a new target domain and the optimality of the learned representation, we will provide some detailed discussion, intuitions, and examples below. We hope that it can clarify things.
> > >
> > > **Regarding the generalizability of the model to a target domain**
> > >
> > > As the reviewer has mentioned, in the paper, we use the motivating example as an example of a source domain and a target domain, to show that even when a model somehow can align the marginal distribution between a source and a target domain, it is not enough. During the rebuttal, since the reviewer asked if StarGAN can successfully learn to transform between these two domains, and since our method only use StarGAN (or any domain translation methods) to transform among the source domains, we treated them as two source domains in the initial rebuttal. The question from the reviewer is now that "If StarGAN can generalize to a new target domain". However, we believe that this question is not very well-posed. The StarGAN model is only trained to transform among the source domains; or in general, with any domain translation method, we will train D\*(D-1) transformations that transform among the D source domains (StarGAN unifies these transformations in a single model by taking the D-dimensional one-hot encoding of the original domain and destination domain of an image as input). Thus, there are no way it can generalize to a new target domain, as it is only trained to learn D\*(D-1) transformations (the transformation is not even defined for a new target domain, since it takes the one-hot encoding vector of size D of the domain, and this one-hot encoding is not defined for a new target domain with index D+1). However, **it does not (and should not) need to generalize to a new target domain**. Its sole purpose is to help to learn an invariant representation among the source domains, and we hope that this invariant property can generalize to a target domain.
> > >
> > > So, a better question is perhaps "can the invariant property generalize from the source domains to a new target domain". Even with this question, the answer is most likely that "we do not have any guarantee". Before we explain further, we would like to emphasize that this question is general for almost all DG methods: most DG methods try to enforce some property of the representation/model across the source domains and hope that they generalize to a new target domain; so the natural question is that can those properties really generalize to the target domain. Therefore, this question and the following discussion are not specific to our method, but are the same for almost all DG methods (all domain-invariance approaches included). As we mentioned, there is no guarantee, since the target domain can technically be anything (we also mentioned this point in our other rebuttal comment regarding your suggestion about an error analysis of the target domain, and in this view, almost all DG methods can be viewed as empirical approaches with strong theoretical motivations -- the guarantee among the source domains). For instance, in the above example, the target domain does not even need to be a circle, it can be a straight line. Even if it is a circle, it can technically have a very large radius (for example, 10000000). The representation network will likely not be able to generalize to these cases or keep any invariant property. However, as we mentioned in the main paper (line 45), these are very extreme examples and should be used for illustrating purposes only. In practice, the datasets are less extreme and the representation network can likely keep some invariant property. For example, if it finds out that the background color is not domain-invariant among the source domains, it can remove this information from the final representation. Let's say a convolutional channel of the final representation corresponds to this information, the model can decide to "turn off" this channel (quite easily by setting the bias of this channel in the final layer to be very negative, which will make the channel value after ReLU to be 0 for all input). Given that the bias is very negative, for a new image from the target domain, this channel value should also be zero. (note that these examples are for illustrating purposes only and it might not happen exactly like this in practice).
> > >
> > > **Regarding the optimality**
> > >
> > > Thank you for your clarification. The reviewer is indeed correct that our method only offers a way to learn a domain invariant representation, and it does not cover the entire search space of all possible domain invariant representations. Regarding this point, we have several thoughts. First of all, as mentioned in our original rebuttal, to the best of our knowledge, most other domain (marginally and conditionally) invariant methods also can only **offer a way to learn an invariant representation**, without searching the entire space. Strong empirical results show that our model outperforms these existing methods, suggesting that our method is "better" than theirs. which is good. Secondly, the reviewer is concerned if "all desired representations do not satisfy our condition". **This cannot happen, and our model is guaranteed to find an invariant representation**. Indeed, there always exist some invariant representations that satisfy the condition in Theorem 1 (a trivial one is that the representation network always returns a constant, this representation is marginally and conditionally invariant and satisfies the conditions in theorem 1; this is a trivial solution but it does guarantee that the scenario the reviewer mentioned cannot happen). Finally, for many optimization problems, it has been shown that narrowing down the search space can be beneficial. For example, the entire search space can be too large for an optimizer/learner, so narrowing it down can be good, as long as we can still find a desirable representation (in this case, the desirable property is domain invariance), especially if the narrowed-down search space has strong and good inductive bias (here we believe that the inductive bias that "the representation network is invariant under domain transformations" is good).

---

> > > > ### Comment · Reviewer_RKBf · 2021-08-26
> > > > **Clarification on "desired model"**
> > > >
> > > > I appreciate the author's instant reply to my question. All your response makes sense to me. I just want to clarify that in my previous discussion, by "desired model", I mean a model which has both 1. domain invariance, 2. discriminative power to the label. In the last paragraph, the author mentions a trivial solution to domain invariance which is "constant representation". However "constant representation" does not preserve the discriminative power of the data to the label. Thus "constant representation" can not be considered as a "desired model".
> > > >
> > > > So my question actually is the following:
> > > > * Is it possible that any desired model (having both **domain invariance** and **discriminative power to the label**) does not satisfy your condition in Theorem 1?
> > > >
> > > > Maybe thinking of this line could derive a new theorem for your work.

---

> > > > > ### Author Response · Authors · 2021-08-27
> > > > > **Discussion (continued)**
> > > > >
> > > > > Dear reviewer,
> > > > >
> > > > > We appreciate your suggestion.
> > > > >
> > > > > As we mentioned, the constant example is just a trivial case to prove that the set of representations that satisfy our condition is not empty, and in practice the method will find a meaningful (discriminative) representation due to the cross-entropy loss.
> > > > >
> > > > > Regarding the question, we can confirm that it is **possible** that there exists a desired representation that does not satisfy the condition in Theorem 1. We can even construct a toy example like that (however, the construction is quite complicated so we will not mention it here). As we mentioned before, our method (and most other domain marginally and conditionally invariant representation methods) only offers a way to find an invariant representation (which is meaningful) without searching the entire search space. Empirical results show that our method outperforms existing ones, which suggests that "our way is better". We can think that there are too many representations that are invariant and discriminative, so even when the search space is narrowed down by our theorem, we are still likely to find one. We can confirm that in practice, almost all methods reach near-perfect performance on the source domains, suggesting that the found representations are discriminative (and if we re-train the prediction head for the target domain while keeping the representation network, we can also get near-perfect performance; therefore these found representations are discriminative for even the target domain, only the invariant property hinders the performance). And as we mentioned earlier, for many optimization/learning problems, narrowing down the search space can be beneficial, especially when we believe that the inductive bias "invariant under domain transformations" is a good inductive bias.
> > > > >
> > > > > We wonder if this answers your question/concern.

---

> > > > > > ### Comment · Reviewer_RKBf · 2021-08-27
> > > > > > **My question is addressed. Increase score to 6.**
> > > > > >
> > > > > > Thank you to the author for the expanded discussion. All my questions are addressed. Through the discussion with the author, it becomes clear that the proposed method is motivated by the belief of "**invariant under domain transformations** is a good inductive bias". Although it has no theoretical guarantee, empirically performance improvement shows the benefit of the proposed method. To acknowledge the author's effort in the rebuttal, I would like to increase my score to 6.

---

> > > > > > > ### Author Response · Authors · 2021-08-27
> > > > > > > **Thank you**
> > > > > > >
> > > > > > > We appreciate your reconsideration!
> > > > > > >
> > > > > > > Best regards,
> > > > > > > Authors.

---

> > > > ### Author Response · Authors · 2021-08-27
> > > > **Discussion (continued)**
> > > >
> > > > Dear reviewer,
> > > >
> > > > We appreciate your suggestion.
> > > >
> > > > As we mentioned, the constant example is just a trivial case to prove that the set of representations that satisfy our condition is not empty, and in practice the method will find a meaningful (discriminative) representation due to the cross-entropy loss.
> > > >
> > > > Regarding the question, we can confirm that it is **possible** that there exists a desired representation that does not satisfy the condition in Theorem 1. We can even construct a toy example like that (however, the construction is quite complicated so we will not mention it here). As we mentioned before, our method (and most other domain marginally and conditionally invariant representation methods) only offers a way to find an invariant representation (which is meaningful) without searching the entire search space. Empirical results show that our method outperforms existing ones, which suggests that "our way is better". We can think that there are too many representations that are invariant and discriminative, so even when the search space is narrowed down by our theorem, we are still likely to find one. We can confirm that in practice, almost all methods reach near-perfect performance on the source domains, suggesting that the found representations are discriminative (and if we re-train the prediction head for the target domain while keeping the representation network, we can also get near-perfect performance; therefore these found representations are discriminative for even the target domain, only the invariant property hinders the performance). And as we mentioned earlier, for many optimization/learning problems, narrowing down the search space can be beneficial, especially when we believe that the inductive bias "invariant under domain transformations" is a good inductive bias.
> > > >
> > > > We wonder if this answers your question/concern.

---

> ### Author Response · Authors · 2021-08-21
> **Regarding the suggestion about theoretical analysis of the model's error**
>
> Dear reviewer,
>
> We also would like to further address your suggestion about a theoretical analysis of the model's error when the domain transformations are not perfectly learned.
>
> Although an analysis of the error on the target domain will strengthen our paper (or any paper in the DG literature), we are afraid that such analysis is not possible with the current domain generalization formulation, since we do not know anything about the target domain and its data. In fact, to the best of our knowledge, most of the papers in the literature (including all baselines mentioned in the paper) also cannot produce such an analysis/bound of the target error. In this aspect, most domain generalization approaches can be viewed as an empirical method (with a strong theoretical motivation).
>
> To clarify, our method learns the domain transformations among the **source domains** only, and enforces the representation to be invariant under those transformations. The algorithm has no access to data of the target domain.
>
> Best regards,
> Authors.

---

### Official Review · Reviewer_qiBT · 2021-07-19

**Rating:** 6
**Confidence:** 4

**Summary:**

This work focuses on solving domain generalization. Given labeled data of multiple source domains, the goal is to predict the labels of target domain data. In domain generalization, the target domain data are not provided at training time. The main challenge of domain generalization is to learn a domain-invariant representation. To solve the problem, most existing approaches focus on the marginal distribution alignment p(z|d) without considering the conditional distribution p(y|z,d). However, it is needed to align both the marginal and the conditional distributions to classify the target data. To address the challenge, they propose a theoretically grounded method for learning a domain-invariant representation based on domain density transformation functions. They enforce the representation to be invariant under the transformations among the source domains to the representation become more robust under other domain transformations. It aligns both the marginal and the conditional distributions. The transformation functions are based on GANs in practice.

**Limitations And Societal Impact:**

- They clearly present the limitations of the proposed method.
- The proposed method is heavily dependent on the transformation network performance. Thus, it requires great effort to set up and train the transformation network.


**Main Review:**

Strengths
- Most existing approaches only focus on the marginal distribution alignment. They point out the limitation of the previous approaches with an eye-catching illustration.
- They propose a novel approach to align both the marginal and the conditional distributions with a theoretical analysis.
- They empirically show the effectiveness of the proposed method by performing experiments on real-world datasets. The proposed method outperforms the relevant baselines.

Weaknesses
- They need to more clarify the differences between the meta learning and domain generalization, and to compare briefly the existing approaches for meta learning and their approach.
- They need to train a complex network (StarGAN) to align the distributions. The classification performance will be greatly affected by training the network.
- They need to explain the reasons for the performance differences with the key ideas of the baselines.


**Time Spent Reviewing:**

10

---

> ### Author Response · Authors · 2021-08-09
> **Rebuttal**
>
> Dear reviewer,
>
> We would like to thank you for your time and effort in helping to review the paper.
>
> **On the difference between meta-learning and domain generalization**
>
> A major difference is that in domain generalization, we typically don't have any data of the target domain, and the model trained on the source domains is kept as is (without any adaptation using the target data) and is expected to perform decently on the target domains. This is also true for our method, no adaptation is allowed for the target domain. The setting of meta-learning also consists of a family of training tasks; however, the model will be given data from the target tasks at test time to "adapt". Therefore, a lot of meta-learning methods (e.g., MAML) aim to learn to "adapt quickly" to new tasks, given a few target data samples and a few parameter updates. As a result, the family of training tasks in meta-learning can be very flexible (they can even be different tasks), while in domain generalization, the task is the same, just that the data distribution may be different for different domains.
>
> **They need to train a complex network (StarGAN) to align the distributions. The classification performance will be greatly affected by training the network.**
>
> In principle, this is correct. However, in practice, we found that our method's performance is not sensitive to the (training of the) transformation network at all. As mentioned in the paper, we use the source code of StarGAN without any modification to the network architecture/hyper-parameters. This shows that our model does not need extensive tuning of the transformation network to achieve good performance. We also experimented on changing the network architecture of StarGAN; however, our performance is not sensitive to these changes.
>
> **They need to explain the reasons for the performance differences with the key ideas of the baselines.**
>
> We believe the performance gain of our method is due to its ability to align the representation distribution better (both marginally and conditionally). These alignments are confirmed in our visualization of the representation space in figure 4.

---

### Author Response · Authors · 2021-08-28
**The discussion period is nearing an end**

Dear reviewers,

Thank you for your time helping to review and improve the paper.

The discussion period is nearing an end, thus we wonder if you can spend some time going over our newest replies, to see if they successfully answered your questions or not. (Reviewer RKBf already finalized their review -- we thank you for that). This is also to give us a decent amount of time to address any of your remaining/additional concerns.

Best regards,

Authors.

---

### Decision · Program_Chairs · 2021-09-27

**Decision:**

Accept (Poster)

**Comment:**

This paper proposes an approach to align both the marginal and the conditional distributions with a theoretical analysis. Authors have done a good job to address many concerns from reviewers. After the discussion period, all three reviewers are inclined to accept the paper (note that one increased the score from 4 to 6 without editing the original review.)